# The human primary visual cortex (V1) encodes the perceived position of static but not moving objects

Man-Ling Ho [1][✉] & D. Samuel Schwarzkopf [1,2]

Brain activity in retinotopic cortex reflects illusory changes in stimulus position. Is this neural signature a general code for apparent position? Here we show that responses in primary visual cortex (V1) are consistent with perception of the Muller-Lyer illusion; however, we found no such signature for another striking illusion, the curveball effect. This demonstrates that V1 does not encode apparent position per se.

[1] UCL Experimental Psychology, 26 Bedford Way, London WC1H 0AP, UK. [2] School of Optometry & Vision Science, University of Auckland, 85 Park Road, Auckland, New Zealand. [✉]email: man-ling.ho.15@ucl.ac.uk

Converging evidence suggests V1 plays a critical role in apparent size representation. Retinotopic activity patterns encode the shift of the object's edges in the hallway illusion[1,2], for veridical depth cues[3], and size adaptation[4]. Importantly, these findings suggest that apparent object boundaries, and hence apparent object size, could be directly read out from the spatial distribution of peak response in V1—an idea that can be traced back to the local sign hypothesis[5,6]. Does V1 encode apparent position shifts in general, regardless of how they arise?

V1 contains a high-resolution retinotopic map of the visual field, which makes it a perfect candidate for a neural representation of object position (and size). Is the retinotopic map in V1 a general reference for the *appearance* of stimulus position? Here we tested this directly using two illusions that modulate apparent position, albeit in very different ways. Our results showed that V1 response is consistent with the perceived position shift in the Muller–Lyer illusion, but not the curveball effect; as such, V1 is unlikely to encode a general reference of apparent position.

## Results and discussion

In our first experiment, we devised a dot variant of the Muller–Lyer illusion, where the distance between two target dots appears further or nearer depending on whether the fins are outward- or inward-facing (Fig. 1a; Supplementary Movie 1: outward-facing, and Supplementary Movie 2: inward-facing). Participants viewed these stimuli while we measured brain activity using functional magnetic resonance imaging (fMRI). Target dots flashed black and white at 2.5 Hz to produce robust V1 responses, while contextual (fin) dots always remained black. Stimuli with outward or inward fins were presented in blocks, each comprising 10 s presentation of contextual dots without target dots (*background-only period*), 16 s of flashing target dots with contextual dots (*illusory period*), followed by a 6 s fixation-only period.

Using population receptive field (pRF) modelling[7] based on another scanning session, we reconstructed the target positions in the Muller–Lyer stimuli based on where V1 responses fell in the retinotopic map (Fig. 1c). Relative to targets with inward fins, the response to targets with outward fins was centred at a location in V1 corresponding to a more peripheral visual field location. We quantified these signatures using a sliding window, collapsed across the hemifields and fit with a Gaussian curve for each individual. The peak location ($\mu$) of the neural signatures should reflect *apparent* shaft length/dot eccentricity

**Fig. 1 Stimuli and group-level data for the Muller–Lyer experiment ($n = 10$). a** Dot-variant Muller–Lyer illusion. The target dots (white) are spaced equidistantly, but with outward/inward fins they appear further apart/closer together. **b** Group-level neural signature collapsed across hemifields and fit with a Gaussian function. The vertical dotted line denotes physical target location. Consistent with the illusion, targets with outward fins appeared more peripheral. **c** V1 responses reconstructed in visual field. Red and blue denote positive and negative responses relative to baseline, respectively. White circles denote the physical target locations. **d** Individual fMRI effects plotted against perceptual effects. The shaded region denotes the 95% confidence interval (CI) estimated through bootstrapping. **e** Predicted neural signatures for simulated target locations (exaggerated for illustrative purpose) were correlated with the measured signatures. **f** Heat map showing correlation coefficient across simulated target locations. The best correlation (red line) for the two conditions was consistent with the illusion measured psychophysically.

(Fig. 1b). As predicted, mean peak eccentricity was greater for outward-fins (mean = 4.49°, std = 0.395) than for inward-fins (mean = 3.98°, std = 0.552; $t(9) = 6.33$, $p<0.001$), in line with the perceived difference in the target location. There were no significant differences for the other Gaussian fit parameters (baseline ($\alpha$): $t(9) = -1.88$, $p = 0.093$; response amplitude ($\beta$): $t(9) = 1.80$, $p = 0.105$; spread ($\sigma$): $t(9) = 1.48$, $p = 0.174$; or goodness-of-fit ($R^2$): $t(9) = -1.20$, $p = 0.260$. These tests were corrected for multiple comparisons with the corrected significance threshold at $0.05/4 = 0.0125$). Mean goodness-of-fit was 0.94 and 0.96 for the outward and inward fin signatures, respectively.

Control analyses showed this was not trivially explained by any systematic differences in eye position between conditions (Supplementary Fig. 1) or residual signals from the context dots. Further, to check whether residual contextual signal contributed to the difference in fit peak location, we applied a more stringent sampling criterion for the sliding window analysis (Supplementary Fig. 2) by imposing a threshold on the maximum pRF size in addition to the location criteria. The goal was to exclude pRFs that overlapped extensively with the context dots. We first applied a lenient size-sampling criterion to exclude vertices whose pRF size (i.e. pRF $\sigma$) exceeded 1° before re-fitting the size signature; this would prevent pRF with extensive overlap with the outer contextual dots. On average, 92% of vertices survived this threshold—although for one participant, there were instances where no pRFs passed the sampling criteria for a particular window (i.e. an empty window). Given it was unclear how this affected the fitting, this participant was excluded from subsequent analyses. The difference in peak location was preserved ($t(8) = 5.55$, $p < 0.001$), and the fMRI effect showed a positive, albeit non-significant, correlation with the perceptual effect ($r = 0.56$, $p = 0.117$, $n = 9$).

Next, we used a highly stringent size sampling criterion, which included only vertices whose pRF size—when taking into account pRF location—did not exceed the height of the sliding window. Given each window extended 0.5° into the upper and lower visual field (i.e. height of 1°), this amounted to selecting vertices where $\text{pRF}y + \text{pRF}\sigma < 0.5°$ (since pRF $\sigma$ is always positive). Here only 35% of the original pRFs (based on location sampling criteria alone) survived thresholding. This excluded pRFs with extensive overlap with all contextual dots. Four participants were excluded from subsequent analyses for having empty windows. The difference in fit peak location was reduced and no longer significant ($t(5) = 2.14$, $p = 0.086$); the correlation between fMRI and perceptual effect was still in the expected direction ($r = 0.43$, $p = 0.394$, $n = 6$). The reduction (but not abolition) of target shift suggests the residual contextual signal at least partially accounts for the fMRI effect. We surmise that because the difference in peak locations of the neural signatures was reduced when restricting the sampling of voxels with larger receptive fields, our findings are most consistent with the spatial pooling theory of the Muller–Lyer illusion[8–10], whereby the illusion arises due to the low-pass filtering of the visual input by cortical neurons.

We also measured the perceived shift in dot location with a psychophysical adjustment task in the scanner. Participants perceived dots with outward fins (mean = 4.18°, std = 0.158) as further apart than dots without fins (mean = 3.99°, std = 0.070), which in turn appeared further apart than dots with inward fins (mean = 3.77°, std = 0.208). This corresponds to an illusion magnitude of ~11%, comparable to previous studies using a dot variant of the Muller–Lyer illusion[11], albeit slightly weaker. The perceptual effect also hinted at a positive correlation with the fMRI effect (outward $\mu-$ inward $\mu$) but this was not statistically significant ($r = 0.51$, $p = 0.133$, $n = 10$; Fig. 1d). Our

reconstruction approach allowed us to further test explicit encoding models. We simulated a set of physical distances between target dots. Then we predicted the reconstructed neural signatures such stimuli would evoke in V1 based on the pRF maps. We correlated sliding window profiles of empirically observed responses (driven by apparent shift) with these predicted responses (based on simulated physical shifts) to determine the physical dot locations that maximally correlated with the observed neural signatures (Fig. 1e). In line with the perceptual effect, the mean predicted location for the outward-fins condition (mean = 4.38°; std = 0.697) was significantly greater than for inward fins (mean = 3.87°; std = 0.649; $t(9) = 6.06$, $p < 0.001$; Fig. 1f).

In this experiment, we focused on V1 because it had been implicated in encoding apparent size in previous studies[1–4,12–16]. We deliberately placed the stimuli on the horizontal meridian so they were centred within the V1 region of each hemisphere and thus ensure optimal coverage by our retinotopic maps. Unlike V1, extrastriate areas V2 and V3 are anatomically separated quadrant field maps. Visual field coverage around the horizontal meridian is therefore sparser, rendering them suboptimal for reconstructing the Muller–Lyer stimuli. Nevertheless, for completeness we also conducted our analysis for V2 and V3 (Supplementary Fig. 3). This revealed a similar pattern of results as in V1, with peak activation being greater for the outward fin condition. Interestingly, the effect was however less pronounced than in V1. Considering the larger receptive field sizes in extrastriate cortex, a simple low-pass filtering of the visual input would predict a greater shift in the peak response than in V1. Moreover, due to the sparse coverage of the horizontal meridian where the target dots were located, the signals in these regions should be dominated by the (physically different) contextual dots. A trivial feedforward account would therefore predict a greater activity shift compared to V1. Instead, our results suggest that the response in V1 is more closely related to the illusory percept. Downstream signals in extrastriate cortex are presumably only inherited from V1.

Position perception is also strongly influenced by visual motion[17]. The curveball illusion[18] is a striking change in the perceived trajectory of a peripheral stimulus induced by a discrepancy in the direction of the stimulus' external motion path and internal texture drift. In our second experiment, participants viewed a variant of this illusion comprising four Gabor patches, each moving vertically towards the horizontal meridian in one visual field quadrant (Fig. 2a; Supplementary Movies 3 and 4). In alternating blocks, the internal motion of the Gabor drifted either orthogonal to the motion path, causing an illusory position shift (Supplementary Movie 3), or along the motion path, a control stimulus without any illusory shift (Supplementary Movie 4). All participants reported a strongly divergent trajectory in the illusion condition. Our quantitative psychophysical experiments revealed a robust illusory trajectory. During debriefing after the scan, most participants drew straight outward trajectories, although a few also reported some curvature. As for the Muller–Lyer experiment, we then used pRF modelling to reconstruct the neural signatures of these stimuli in V1. This revealed a clear response signature of the veridical motion paths in both conditions (Fig. 2c).

To maximise sensitivity, we collapsed data from the four quadrants. We again sampled data using a sliding window and fit these signatures with a Gaussian curve for each participant. But here, we repeated this procedure in two locations corresponding to the start and end of the motion path, respectively (Fig. 2b). The illusion should manifest in a peripheral shift in the peak location of the neural signature. Therefore, we expected a main effect of condition and an interaction between the condition and sliding

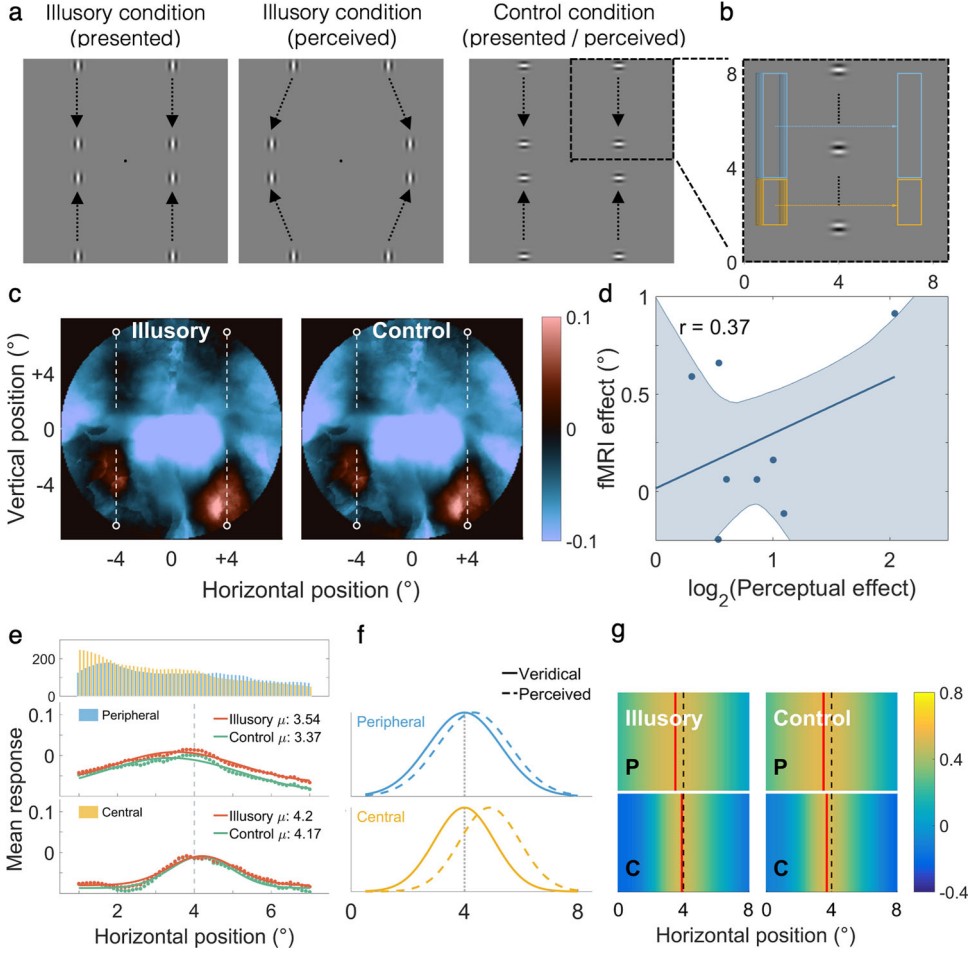

**Fig. 2 Stimuli and group-level data for the Curveball experiment (*n* = 10). a** Gabor patterns in the illusory and control conditions travelled the same physical path. However, in the illusory condition, the internal gratings drifted in the direction orthogonal to the external path (towards the periphery); causing the perception of a 'curveball' when viewed in the periphery. **b** Sliding window sampling was used at the peripheral (blue) and central (yellow) location, corresponding to the start and the end of the motion path, respectively. Window height accounted for cortical magnification. **c** V1 responses reconstructed in visual field. Red and blue denote positive and negative responses relative to baseline, respectively. White dashed lines denote the physical motion paths. **d** Individual fMRI effects (defined as illusory μ– control μ in the central location) plotted against perceptual effects. The shaded region denotes the 95% CI estimated through bootstrapping. **e** Group-level neural signatures collapsed across hemifields and fit with a Gaussian function. Both illusory and control signatures across both sliding window locations were centred on the physical motion path (vertical dotted lines). **f** Predicted neural signatures for simulated physical motion trajectories were correlated with the measured signatures. **g** Heat map showing correlation coefficient across various simulated physical motion paths. The best correlated (red line) signatures corresponded to the physical motion path.

window location, where the shift should be maximised towards the end-of-motion path in the illusory condition (Fig. 2f). Neural signatures with especially poor fits (defined as $R^2$ value 2.5 standard deviations below mean) or with highly dissimilar goodness-of-fit values between conditions (where difference in goodness-of-fit exceeded 0.3) were excluded. Following these exclusion criteria, S3 and S5 were removed from the analysis. While the mean peak location of the signatures was subtly shifted in the expected direction (Fig. 2e), this difference was not statistically significant, and there was no interaction between condition and sliding window location. No difference was found for other fit parameters either, including baseline, response amplitude, spread or goodness-of-fit (Supplementary Table 1). Mean goodness-of-fit was 0.86 and 0.87 for the illusory and control signatures in the peripheral location, and 0.89 and 0.88 in the central location, respectively.

The shift in peak location of the signatures towards the end of the motion path also did not correlate with perceptual effect (Fig. 2d). There were no systematic differences in eye movements between conditions (see Supplementary Fig. 4). We observed

similar results in V2 and V3 (Supplementary Fig. 5). As for the Muller–Lyer illusion, we used our encoding model to predict sliding window profiles of a range of simulated physical motion paths. The simulated motion path that best matched the observed signature was the veridical path (Fig. 2g).

Taken together, our results reveal a neural signature of the apparent position shift in the Muller–Lyer illusion in V1. This effect seems less compatible with misapplied size constancy scaling theory[19] and more consistent with the idea that position distortion arises from a centroid extraction process, whereby peak response reflects spatial pooling of contextual signals in the periphery[8–10]. Previous work has shown that the Muller–Lyer illusion interacts with viewing distance[8] and that adaptation to low spatial frequency gratings reduces illusion strength[20]; both findings are consistent with the interpretation that this illusion depends on neurons with large receptive fields. Our results further support the spatial pooling account because in line with larger peripheral receptive fields outward fins exert a stronger outward 'pull', while the peak response in the inward-fins condition centred around the veridical target location.

We note that the strength of the Muller–Lyer illusion in our experiments was weaker than in many psychophysical reports. Such differences may relate to stimulus differences: Psychophysical experiments usually present the stimuli close to the centre of gaze. We necessarily placed our stimuli in the parafovea to produce robust and discriminable retinotopic activations. Further, we used a dot variant of the Muller–Lyer illusion, which has been shown to be weaker than line stimuli[11]. Estimates of illusion magnitude may also depend on how the illusion is measured: Many previous studies used a line-bisection task[20] whereas we used a position matching task. These were deliberate experimental choices: Dot stimuli are essential for producing isolated response clusters in V1. Similarly, it is imperative that we match the stimulus conditions in our illusion and reference stimuli as much as possible. One previous study[21] reported a weaker illusion when observers adjusted the location of end points (like in our task) compared to judging the length of lines. Line stimuli suffer from the same issue with localisation, however—the cortical activation pattern evoked by a line stimulus is necessarily different from that evoked by isolated dots. As such, matching the length of a line to the separation of our target dots is likely contaminated by additional processing of this line reference. In any case, the estimate of any perceptual effect must always be the product of the whole brain working in concert. It seems therefore likely that multiple separate factors give rise to the Muller–Lyer percept. The V1 signals may only relate to one factor, presumably related the low-level processing, but it may not correlate with other, higher-level factors contributing to the illusion.

In contrast to the Muller–Lyer illusion, we found no evidence that V1 carries a signal reflecting the perceived motion path in the curveball illusion. This is consistent with converging evidence from multivariate decoding analysis[22]. However, decoding analysis is inherently opportunistic and agnostic to how a percept is encoded[23]. Particularly in early visual areas decoding is prone to eye movement confounds[24], and it depends on arbitrary experimental choices such as the number of voxels to include in the multivariate pattern. Our model-based reconstructions instead test explicit assumptions about how the brain encodes a percept. We demonstrate that V1 does not encode the motion-induced shift in trajectory. Interestingly, other kinds of motion-induced position shifts show correlates in early visual cortex[25], albeit not necessarily in the direction predicted by perceptual shifts[26,27]. These findings suggest that in specific situations V1 response modulation may shift activity peaks, in turn producing downstream representations in higher brain regions. The visual system then interprets this as position (or size) differences. Such V1 modulation may occur through local interactions, and our results suggest the Muller–Lyer illusion is caused by such a process. Other likely candidates for such local interactions are the Ebbinghaus and Delboeuf illusions[14,15,28], and possibly the effect of size adaptation[4]. Conversely, feedback from higher areas could also modulate V1 activity; this seems a more plausible explanation for size illusions produced by depth cues[2,12,13]. However, our findings provide crucial evidence that the high-resolution retinotopic map in V1 does not constitute a general reference for apparent stimulus position.

## Methods

**Participants**. All participants had normal or corrected-to-normal vision and provided informed, written consent prior to participation. The study was approved by the University College London Research Ethics Committee. Ten participants (five females; age range 23–49 years; two left-handed) took part in the Muller–Lyer experiment, including one of the authors. Ten participants (two from the Muller–Lyer experiment; seven females; age range 20–48 years; two left-handed) took part in the Curveball experiment.

**Stimuli and tasks**. All stimuli were generated using MATLAB R2014a (Version 8.3; The MathWorks Inc., 2014) and Psychtoolbox (Version 3.0.11; Brainard, 1997) and projected onto a screen (36.8 × 20.2 cm; resolution 1920 × 1080 pixels) at the back of the scanner bore and were viewed through a mirror mounted on the head coil at a distance of 67 cm, resulting in a screen size of 30.7° × 17.1°. In all experiments, the order of the stimulus conditions was pseudo-randomised without replacement. Eye movements were monitored using an MR-compatible SR Research EyeLink 1000 eye tracker with data sampled at 60 Hz.

*Retinotopic mapping*. The mapping procedure involved the simultaneous presentation of a rotating wedge and an expanding–contracting ring[29]. The mapping stimuli were 228 coloured natural images and their phase-scrambled versions. The intact images were of cityscapes, outdoor sceneries, animals, faces, textures and written scripts. One image contained an Anderson tartan pattern—the target in an image detection task. The images were scaled to the height of the screen and cropped into circles (diameter of 17.03°) with the remainder of the screen filled with grey. The images were viewed through the combined ring-and-wedge aperture, with image order pseudo-randomised within each run. Image display changed every 500 ms.

The wedge aperture (12° polar angle) rotated around a black central fixation dot either clockwise or counter-clockwise in 60 discrete steps (1 step/s) with a 50% overlap with consecutive wedges. A ring aperture expanded or contracted in 36 logarithmic steps (1 step/s) while maintaining a proportional annulus dimension such that the diameter of the inner circle (minimum of 0.48°) was always 56–58% of the outer circle with an 89–90% overlap with consecutive rings. The maximal eccentricity for the apertures was 8.5°. Each mapping run had a total of three cycles of wedge rotation and five cycles of ring expansion–contraction. Each block began with 90 s of stimulus presentation followed by a 30 s fixation-only interval. Within each block, image type alternated between intact and phase-scrambled every 15 s.

Participants were asked to fixate on the central fixation dot (diameter of 0.13°) at all times. Every 200 ms, there was a 0.03 probability of the fixation dot changing from black to either red, green, blue, cyan, magenta, yellow, white or black for 200 ms. Participants were asked to press a button on a response box whenever the fixation dot turned red or whenever the Anderson tartan pattern appeared. To aid fixation, a low contrast polar grid (line width of 0.02°; opacity of 10.2%) centred around the fixation dot was superimposed onto the foreground at all times. This consisted of 10 circles with radii evenly spaced between 0.38° and 27.35°, and 12 evenly spaced radial lines (lines at every 30° polar angle).

*Muller–Lyer experiment*. The target stimuli were two black dots (diameter of 0.64°) equidistant from a white fixation dot (diameter of 0.16°) positioned at the centre of the screen with the rest of the screen filled with grey. The centre-to-centre distance between the target dots and the fixation dot was 4°. Eight contextual dots were positioned at a 45° polar angle from the target dots such that they formed two inward or two outward fins, depending on the condition. The centre-to-centre distance between a target and contextual dot, or between two contextual dots was 1.36°. The target dots flashed between black and white at 2.5 Hz, while the contextual dots remained black at all times.

Stimuli were presented in blocks each comprising 10 s presentation of contextual dots without target dots (background-only period), 16 s of flashing target dots with contextual dots (illusory period), followed by a 6 s fixation-only period. Each run contained a total of eight blocks, where each condition appeared four times. Participants completed ten runs of the experiment. The fixation dot was positioned at the centre of the screen throughout the run. To ensure fixation, two tiny circles (diameter: 0.02°) appeared around the fixation dot (probability of 0.001 at every refresh) such that it would form the percept of a Mickey Mouse logo in four possible orientations for a period of 167 ms. The target dots occasionally flashed to either red, green, blue or yellow (probability of 0.0025 every 200 ms). Participants had to press a button whenever an upright Mickey Mouse logo formed around the fixation or whenever the target dots turned red.

After the main fMRI experiment, participants completed 48 trials of an adjustment task in the scanner. The reference stimuli were either target dots with outward fins, inward fins or without any context. The test stimuli were always white and without context. The test and reference stimuli were placed side-by-side on-screen. There were six trial types (3 references × 2 reference/test location). Participants were asked to adjust the distance between the test dots until it matched the distance in the reference. The distance between test dots (i.e. shaft length) was varied as a binary logarithm of the size ratio of test over reference with variation drawn from a Gaussian distribution with $\mu = 1$ and $\sigma = 0.25$. Participants had unlimited viewing time and were asked to freely scan with their eyes between the stimuli during adjustment.

*Curveball experiment*. Four Gabor patterns (sinusoidal gratings within a Gaussian envelope) with a wavelength of 0.4° and 100% contrast were presented on a uniform grey background. The standard deviation of the Gaussian envelope was 0.192°. A black fixation dot (0.13°) was presented at the centre of the screen. One Gabor pattern was placed in each visual quadrant such that their positions were mirrored across the vertical and horizontal meridian. The four Gabor patterns travelled simultaneously and vertically at a speed of 2.88°/s (external motion) for 2.5 s from the edge of the screen towards the horizontal meridian, with a horizontal

position of 4° from the vertical meridian, covering a distance of 7.2° (no reversal). The internal motion had a temporal frequency of 10 Hz and drifted either orthogonally and outward relative to the motion path (illusory stimulus) or in the direction opposite to the motion path (control stimulus).

Stimuli were presented in blocks. Each run started with an initial 15 s fixation-only period, followed by six repeats of a 15 s stimulus period, a 1 s fixation-only interval, another 15 s stimulus period and a final 15 s fixation-only period. During the stimulus periods, either the illusory or the control stimulus was presented, where the Gabor patterns travelled their path a total of six times (2.5 s per trip). Participants completed eight runs of the main fMRI experiment. To ensure fixation, the fixation dot occasionally flashed to red, green or blue for 333 ms (probability of 0.001 every refresh, or 0.0167 s). Participants were asked to fixate on the fixation dot at all times and to press a button when the fixation dot turned red, or when the spatial frequency of the Gabor patterns decreased to a wavelength of 0.32° for the duration of a trip (probability of 0.1 per trip).

After the main fMRI experiment, the perceptual effect was measured inside the scanner (10 trials per condition). Participants adjusted the distance between two squares ($0.2° \times 0.2°$) along the horizontal meridian to match the last seen horizontal position of the stimuli. The initial position of the adjustment squares was centred on the physical motion path (i.e. 4° eccentricity) and varied based on a Gaussian distribution of $\mu = 1$ and $\sigma = 0.1$. To avoid adaptation, the squares flashed between red and light grey at a rate of 2.5 Hz. To prevent participants from using the adjustment squares as position references, the squares were hidden during stimulus presentation and were only made visible 800 ms after stimulus offset. There was no time limit, and participants had the option of replaying the stimulus motion. Given the adjustment task only captured the last seen position of the stimuli, participants also indicated by drawing on a piece of paper, the perceived motion path of the stimuli in the two conditions outside the scanner.

**MRI data acquisition**. All functional and anatomical images were acquired on a Siemens Avanto 1.5 T MRI scanner with a customised 30-channel head coil (32-channel with two anterior channels removed to avoid restriction of view). Functional images were collected using T2*-weighted multi-band 2D echo-planar imaging sequence[30] centred around the occipital cortex (TR = 1000 ms, TE = 55 ms, flip angle = 75°, 36 transverse slices, acceleration factor = 4, FOV = 96 × 96 voxels) at a resolution of 2.3 mm isotropic voxels. Slices were tilted to be approximately parallel to the calcarine sulcus to ensure coverage of the occipital cortex, and the occipital–temporal and inferior parietal cortices.

Functional images for the retinotopic mapping procedure were acquired across three runs (490 volumes per run). For the Muller–Lyer experiment, 266 volumes were collected per run, with a total of 10 runs. For the Curveball experiment, 301 volumes were collected per run, with a total of eight runs. The first ten volumes of each run were discarded to allow for the fMRI signal to reach equilibrium. Data for retinotopic mapping and the main fMRI experiment were acquired in separate sessions; this was not considered an issue given that pRF estimates are stable across sessions[31–33].

A high-resolution anatomical image was acquired per participant using T1-weighted, magnetisation-prepared rapid acquisition with gradient echo (MPRAGE) sequence (TR = 2730 ms, TE = 3.57 ms, 176 sagittal slices, FOV = 256 × 256 voxels) at a resolution of 1 mm isotropic voxels. As the high-resolution anatomical scan was collected in the same session as the retinotopic mapping, an additional fast MPRAGE scan was collected after the main fMRI experiment to aid co-registration (TR = 1150 ms, TE = 3.6 ms, 80 sagittal slices).

**Preprocessing**. All functional images were preprocessed with SPM12 using default parameters (Version 6685; Wellcome Trust Centre for NeuroImaging). The images were bias-corrected for intensity inhomogeneities, realigned, unwarped and co-registered to the high-resolution anatomical scan. The fMRI time series for each voxel were linearly detrended and z-score normalised; these were averaged across runs for the retinotopic mapping procedure and concatenated across runs for the main experiment. Functional data were projected onto a 3D reconstruction of cortical surfaces using FreeSurfer[34,35] (Version 5.3) by finding for each vertex in the surface mesh, the corresponding voxel in the functional images falling at the medial position between the grey-white matter boundary and the pial surface. All subsequent analyses were done in surface space, including only vertices in the occipital lobe.

*pRF modelling*. We modelled the pRF location $(x, y)$ and size $(\sigma)$ for each vertex as a two-dimensional Gaussian function in a two-stage procedure[7] (SamSrf Toolbox Version 5.84 for pRF analysis; https://osf.io/mrzqy/). In the *coarse-fitting* step, we first generated pRF profiles through an extensive grid search (15 $x$ values × 15 $y$ values × 34 $\sigma$ values), with $x$ and $y$ values stepping evenly from −8.925° to 8.925°, and $\sigma$ values stepping logarithmically from 0.18° to 17°. A predicted time series was generated for each parameter combination by calculating the overlap between the pRF profile and a binary mask (100 × 100 pixels) corresponding to the mapping stimulus. These were convolved with a canonical haemodynamic response function (HRF) obtained in a previous study[36]. Pearson correlation was calculated between each predicted time series and the observed time series for data that had been spatially smoothed on the spherical surface mesh with a kernel (full-width at half maximum, FWHM) of 5 mm. The parameters that produced maximal correlation

while surviving the goodness-of-fit threshold ($R^2 > 0.05$) were entered into a slower *fine-fitting* step. Here the three pRF parameters obtained from the coarse fit were used as seed values for further optimisation, which involved minimising the sum of squared errors between the predicted and the unsmoothed observed time series. The optimisation also included a fourth response amplitude ($\beta$) parameter. The final parameter maps were smoothed across the spherical surface with a kernel (FWHM) of 3 mm and projected onto a spherical model of each hemisphere for visualisation and delineation of visual areas.

*Delineation*. Separately for each cortical hemisphere, V1–V3 were delineated based on smoothed polar angle, eccentricity and field sign maps using the SamSrf toolbox. The reversals of the polar angle indicated the boundaries between visual areas[37]. V1 was delineated as full hemifield maps within the calcarine sulcus. V2v, V2d and V3v, V3d encircling V1 were delineated as quarter-field maps and merged into V2 and V3, each containing full hemifield representations. While V3A was also visible in most hemispheres, the visual field coverage in this area was often incomplete and could not always be clearly delineated. V3A comprises only a small cortical territory with large, peripherally-biased pRFs. This made it impossible to generate accurate reconstructions of the stimuli from this region.

*Block design analysis*. The concatenated time series of the main fMRI experiments were entered into a GLM using SamSrf. Boxcar regressors were defined per condition and convolved with the canonical HRF[36]. The GLM further included six motion regressors and a global covariate. In the Muller–Lyer experiment, signal for the target dots was isolated by contrasting background-only periods with illusory periods ('outward', 'inward'). In the Curveball experiment, fixation-only periods were subtracted from the stimulus periods ('illusory', 'control'). This was followed by smoothing across the spherical surface mesh (FWHM of 3 mm) and selection of vertices surviving a goodness-of-fit threshold ($R^2 > 0.05$) in the pRF model.

*Sliding window reconstruction*. Based on pRF parameters, it was possible to identify and sample vertices whose pRF fall within the target location in a step-wise fashion, giving rise to a quantifiable neural signature. For the Muller–Lyer stimuli, a 1° × 1° window stepped across the horizontal meridian at a step size of 0.1°, where responses of vertices falling within the window were sampled and averaged. Responses between 1.5° and 6.5° eccentricity were collapsed across the left and right hemifields and fit with a one-dimensional Gaussian function with four parameters: baseline ($\alpha$), response amplitude ($\beta$), peak location ($\mu$) and spread ($\sigma$).

For the curveball stimuli, responses were sampled in two sliding window locations to capture target representation at the start and end of the motion path. At both locations, responses of vertices were sampled and averaged using a 'tall' window (peripheral: 1° × 4.5°; central: 1° × 2°) that stepped horizontally from 1° to 7° at a step size of 0.1°. The combined vertical coverage of the windows corresponded to the length of the motion path. Responses were sampled by collapsing across the quadrants and fit with a one-dimensional Gaussian function.

*Response prediction reconstruction*. For a more 'model-free' reconstruction approach, we also compared response profiles of apparent position shift with predicted responses to simulated physical shifts. Here predicted responses were obtained by overlaying pRF profiles with binary masks corresponding to various hypothetical stimuli positions. In the Muller–Lyer experiment, we generated a total of 130 binary masks corresponding to various simulated dot locations (distance increased by 0.128° per mask, growing symmetrically across the left and right hemifield). In the Curveball experiment, we generated 81 binary masks corresponding to hypothetical physical motions path at various angles. Note that each mask was generated by averaging across 150 frames (corresponding to 150 positions). We rectified the original image matrix by applying the following: $|img - 0.5| * 2$. This effectively sets the background to black, inverts the dark troughs of the sinusoidal gratings, and normalises the image matrix to have a maximum value of 1. For both experiments, the sliding window profiles for the predicted responses were then correlated with the observed responses to determine the physical position shifts that best matched the apparent shift based on observed response. Vertices with pRF falling outside of the screen dimension (8.5°) were excluded from all reconstruction analyses.

*Backprojection for visualisation*. For both experiments, neural responses in the different conditions could be back-projected into visual space for visualisation (SamSrf Toolbox Version 6.19 for back-projection). Using a searchlight summary procedure[29], a grid was first generated to cover the visual space with a 0.1° horizontal and vertical spacing between grid points. A 1° radius 'searchlight' centring around each grid point was used to sample and average responses from all vertices with pRF centres falling within the searchlight. Each searchlight was weighted based on the count and inverse distance of the pRFs from the searchlight centre; this was reflected in the saturation level of the heat map. The responses were averaged across participants to reconstruct apparent size perception at the group level[38].

*Behavioural task*. For the Muller–Lyer experiment, the perceptual effect for each context was calculated by log transforming the linear size ratio of the adjusted distance

over the reference distance. The separate effects for the outward- and inward-fins context were summed (i.e. outward size ratio–inward size ratio, where the inward size ratio should be negative) to find the total perceptual effect. For the Curveball experiment, the magnitude of the perceptual effect was calculated by subtracting the mean position of the illusory trials from the mean position of the control trials.

*Eyetracking analysis.* Slow drifts in the eyetracking time series were detrended using a 'model-free' sliding window approach[29]. Here we defined a window (width of 10 s) that stepped across the eyetracking time series (step size of 1 s) where we calculated the median of all position samples falling within a given window. The vector of 'local medians' were then subtracted from the original time series to obtain the detrended time series. Eye position data were then pooled across runs for each condition and visualised as a two-dimensional histogram (bin size of 0.05°) and normalised to have a maximum value of 1. Eye position variability was estimated using the median absolute deviation.

**Statistics and reproducibility**. Statistical analyses were performed using paired sample *t*-tests to compare neural and behavioural data between the illusory and control conditions within subject. The analyses have been described in the main text and the corresponding "Methods" sections to allow reproducibility.

**Reporting summary**. Further information on research design is available in the Nature Research Reporting Summary linked to this article.

## Data availability
Preprocessed data for generating the plots shown in this article are publicly available at https://osf.io/5qxab[39].

## Code availability
MATLAB scripts for generating the plots shown in this article are publicly available at https://osf.io/5qxab[39].

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

## Author contributions
M.H. and D.S.S. co-conceived and designed the studies. M.H. performed experiments, analysed the data, prepared the figures, and wrote the methodology. D.S.S. supervised the studies, acquired funding, and wrote the main manuscript.

## Competing interests
The authors declare no competing interests.
