## [Peer Review File · Communications Biology]

Reviewers' comments:

Reviewer #1 (Remarks to the Author):

Across two experiments participants were presented visual illusions that induce position shifts while BOLD activity was recorded. Separately, the population receptive fields of V1 voxels were measured. It is reported that population receptive fields respond according to the perceived position of the Müller-Lyer illusion, but according to the veridical position of the curveball illusion. It is concluded that V1 does not encode perceived position. This interpretation is consistent with and an extension of previous work showing that the position of the curveball illusion could not be decoded from V1. The paper is well written, and I think it is a relevant contribution. However, I have a few issues which I think should be addressed:

1. In the methods section you say that you delineated not only V1, but also V2 and V3. Is there a reason further analyses of V2 and V3 were not done / reported? There is behavioral data from Cavanagh and Tse (2019) that shows it is unlikely for V1 to be the source of the curveball illusion (they call it double-drift illusion, but it is essentially the same thing). They also suggest it might be V3a, so you could potentially test that. <https://cpb-us-e1.wpmucdn.com/sites.dartmouth.edu/dist/9/2172/files/2020/01/85.pdf>

2. Given the behavioral data from Cavanagh & Tse (2019) and the decoding results that you cited that rule out V1 as the origin of the illusion, your results for the curveball illusion make sense. However, I'm wondering how strong the curveball illusion was with your stimulus configuration, because the way you had set it up, there can be many "resets" of the illusion back to the physical position (see for example 't Hart, B., Henriques, D. Y., & Cavanagh, P. (2021). Measuring the double-drift illusion with hand tracking. <https://www.biorxiv.org/content/10.1101/2021.08.06.455415v1.full.pdf>). Your Gabors moved for 2.5 seconds, and resets can occur after 1.1 seconds. Your Gabors moved over 7.2 dva, and resets can occur around 3.6 dva. If too many of these resets occurred, then the perceived position might have been the physical position, which could also explain why the population receptive fields don't show an effect of the illusion. Can you provide more details on the perceptual effect to show that the perceptual effect was not diminished by resets? You made participants draw the stimulus, did they all draw straight lines for the illusion trajectory? A side note to this: Where you correlate the perceptual effect with the fMRI effect (figure 1D and figure 2D), what are the units of the perceptual effects?

3. I would appreciate a more detailed demonstration of the illusions. Since dynamic displays were used for both illusions, videos might be appropriate. However, schematics like the one showing the curveball effect (figure 2A) are also fine. Figure 1A is supposedly showing the Müller-Lyer stimulus, but it seems to show data instead.

Reviewer #2 (Remarks to the Author):

This paper shows that activity on V1 does reflect the shifted, perceived locations of dots influenced the Müller-Lyer illusion. In contrast, the deviation of the perceived motion path of the curveball illusion does not affect V1 activity which instead follows the physical path.

The conclusion is that therefore V1 does not encode position per se. This is somewhat nebulous as a main conclusion and something that was already in the literature (e.g., Whitney et al., 2003; Liu et al, 2019).

There are some issues with the Müller-Lyer experiment. The stimulus is in dot format but there is no image of the test stimulus that can allow readers to judge how much illusion they see in this

format. The methods describe a figure that is only fins with no elements along the space between the two vertices. Eight contextual dots are mentioned, I assume that is 4 on each fin, making 5 dots in total for the fins and vertex on one end and another 5 at the other end, for 10 dots total. The authors report a 10% shift in length between the fins out and fins in versions in their behavioral test. This is a small illusion for the Müller-Lyer. In 1970, Coren reported that the illusion effect was about 20% for a line version and 15% for a dot version (with 2 dots to make each fin, so 6 dots in all). Both are larger effects than those reported by the authors here.

Part of that may be due to the measurement of the endpoints rather than some size matching. Gillam and Chambers (1985) report very little Müller-Lyer effect when they had subjects adjusted dots to line up with the endpoint locations. The effect was six times larger when the subjects matched perceived length with an adjustable comparison line. Reduced illusion for the line endings is consistent with the result that reaching to the vertices is quite accurate (Cook, 1979; Thompson & Westwood, 2007).

Finally, the recovered location of the vertex might show an influence of the responses to the contextual dots biasing them inward for the fins in condition and vice versa. This is just the low-pass filter model of the effect but now applied to the cortical patterns of activity. I don't see how the authors have controlled for this. I may have missed something.

For the curveball illusion, the absence of an effect in V1 has already been reported (Liu et al, 2019).

Reviewer #1

Across two experiments participants were presented visual illusions that induce position shifts while BOLD activity was recorded. Separately, the population receptive fields of V1 voxels were measured. It is reported that population receptive fields respond according to the perceived position of the Müller-Lyer illusion, but according to the veridical position of the curveball illusion. It is concluded that V1 does not encode perceived position. This interpretation is consistent with and an extension of previous work showing that the position of the curveball illusion could not be decoded from V1. The paper is well written, and I think it is a relevant contribution. However, I have a few issues which I think should be addressed:

We thank the reviewer for this succinct summary and positive comments on our study.

1. In the methods section you say that you delineated not only V1, but also V2 and V3. Is there a reason further analyses of V2 and V3 were not done / reported? There is behavioral data from Cavanagh and Tse (2019) that shows it is unlikely for V1 to be the source of the curveball illusion (they call it double-drift illusion, but it is essentially the same thing). They also suggest it might be V3a, so you could potentially test that. <https://cpb-us-e1.wpmucdn.com/sites.dartmouth.edu/dist/9/2172/files/2020/01/85.pdf>

Thank you for raising this point. We primarily focused on the results from V1, because previous research indicated that V1 encodes¹⁻⁶ and correlates with⁷⁻⁹ the perceived size of visual objects. Moreover, as V1 is the largest of the early retinotopic cortical regions with the smallest receptive field sizes it is the primary candidate for a potential “workspace” mediating such fine spatial detail. Therefore, we regard results from the extrastriate regions secondary to the aims of our study.

Further, the design of our Muller-Lyer experiment was suboptimal for analyzing data from V2 and V3: we deliberately placed the stimuli on the horizontal meridian to ensure they were centered within the V1 region, thus maximizing the cortical territory encoding the stimuli in that region. V2 and V3 are quadrant field maps, which results in data loss around the horizontal meridian and so the reconstructions of brain activity in that experiment will be less reliable than in V1.

However, these points of course do not preclude us from conducting these analyses. This visual field coverage issue also does not apply to the curveball experiments where stimuli were placed in each quadrant. We therefore now include equivalent plots for these regions. The results from V2 and V3 are consistent with those from V1. Interestingly, the magnitude of the Muller-Lyer effect is smaller than in V1, suggesting that the activation pattern in V1 is indeed more relevant for the perceptual illusion. The responses of V2 and V3 are effectively what would be expected as those regions with

their larger receptive fields sample from the responses in V1. We now added this tentative point in our Discussion.

Regarding V3A, this is a relatively small region with large population receptive fields and a considerable peripheral bias. The map quality for delineating this region reliably was also less clear in some participants. This drastically limits the usefulness of the maps for this reconstruction analysis. Due to these issues, we decided not to include this in our revised manuscript, unless reviewers and editors consider this crucial.

2. Given the behavioral data from Cavanagh & Tse (2019) and the decoding results that you cited that rule out V1 as the origin of the illusion, your results for the curveball illusion make sense.

However, I'm wondering how strong the curveball illusion was with your stimulus configuration, because the way you had set it up, there can be many "resets" of the illusion back to the physical position (see for example 't Hart, B., Henriques, D. Y., & Cavanagh, P. (2021). Measuring the double-drift illusion with hand tracking.

<https://www.biorxiv.org/content/10.1101/2021.08.06.455415v1.full.pdf>). Your Gabors moved for 2.5 seconds, and resets can occur after 1.1 seconds. Your Gabors moved over 7.2 dva, and resets can occur around 3.6 dva. If too many of these resets occurred, then the perceived position might have been the physical position, which could also explain why the population receptive fields don't show an effect of the illusion. Can you provide more details on the perceptual effect to show that the perceptual effect was not diminished by resets?

Via extensive piloting prior to the experiments, we designed our stimuli to produce pronounced illusory effects in both experiments. Our participants reported strong illusory effects for the curveball illusion without such "resets". During debriefing all participants reported a stable trajectory for these stimuli. To give readers a better impression of the stimuli, we now include an example video of the curveball stimuli.

You made participants draw the stimulus, did they all draw straight lines for the illusion trajectory?

All participants draw a trajectory diverging notably from the control condition. Most participants reported a straight trajectory (see left image below) moving peripherally although a few also reported a slight curvature to the path (see right image for one example). All participants were rather consistent about the perceived end point of the trajectory, however.

Due to relative vagueness of these drawings and their *posthoc* nature after the scan, we consider the empirical psychophysical data collected inside the scanner to be more useful. We therefore decided not to include these drawings in the revised manuscript but only briefly discuss the general findings. However, if the reviewer feels these drawings should be included, we could be convinced to do so:

A side note to this: Where you correlate the perceptual effect with the fMRI effect (figure 1D and figure 2D), what are the units of the perceptual effects?

We apologize for the lack of detail here. The units were binary log units. Specifically, the illusion was determined by calculating the ratio of the shaft length (i.e. distance between target dots) in each the outward and inward fins conditions relative to the true separation. This ratio was transformed into a binary logarithm and the perceptual effect used in our analysis is the difference between the log values for the two conditions.

3. I would appreciate a more detailed demonstration of the illusions. Since dynamic displays were used for both illusions, videos might be appropriate. However, schematics like the one showing the curveball effect (figure 2A) are also fine. Figure 1A is supposedly showing the Müller-Lyer stimulus, but it seems to show data instead.

This is an important point, and we agree that a demonstration of the illusions is useful here. We now included animations for both illusions. We also apologize for uploading an incorrect version of Figure 1; this was indeed supposed to be a static illustration of the Muller-Lyer stimulus. This has now been corrected in our revision.

Reviewer #2

This paper shows that activity on V1 does reflect the shifted, perceived locations of dots influenced the Müller-Lyer illusion. In contrast, the deviation of the perceived motion path of the curveball illusion does not affect V1 activity which instead follows the physical path.

The conclusion is that therefore V1 does not encode position per se. This is somewhat nebulous as a main conclusion and something that was already in the literature (e.g., Whitney et al., 2003; Liu et al, 2019).

We thank the reviewer for their constructive feedback. We respectfully disagree that this conclusion is “nebulous”, especially since this statement is directly contradicted by the latter half of this sentence. Several studies¹⁻⁶ indicated that V1 responses encode perceived stimulus size (which is confounded with perceived position). This raises the question whether any perceived change in stimulus position could be encoded in this region. Our study now directly compares this using a visual size illusion not previously used in this context (Muller-Lyer) and a position-shift illusion (curveball). In this direct juxtaposition, we confirm a neural correlate in V1 for the Muller-Lyer percept but not the position-shift induced by the curveball illusion.

It is true that Liu et al¹⁰ reported a similar result for the curveball illusion and we already discussed this in detail previously. However, their study used decoding analysis which is inherently agnostic to what underlies voxel response patterns. While the study by Liu et al is important because it suggests that the perceived motion trajectory is represented in frontal regions, it cannot specifically rule out the existence of a representation in early visual cortex. Any number of confounding factors can obscure successful cross-classification in decoding analysis: performance of classifiers is inherently dependent on the variability of response patterns and several experimental choices, such as the number of voxels included in the analysis, which will differ across brain regions. Critically, it does not allow any inference about whether the brain activations are biologically plausible. Our reconstruction approach enables us to directly test models of the underlying representation. We show that the activation caused by our curveball stimuli in early visual areas is consistent with what is predicted from retinotopic location alone. We argue that both ours and Liu et al’s experiments provide crucial convergent evidence on this effect.

The other study mentioned by the reviewer, Whitney et al 2003¹¹, in fact reported the exact opposite: it showed that signals in V1 reflect the perceived position of drifting Gabor stimuli (whose position remains static). Extrapolating from this finding to the curveball illusion would therefore

predict a V1 representation of the perceived position shift in that illusion as well – our results and those by Liu et al¹⁰ refute that possibility conclusively. Interestingly, the work by Whitney also suggests the possibility that V1 signals encode positions trailing the perceived location^{12,13}; our results also show no evidence of that suggesting differences between the curveball effect and the static stimuli used in those earlier experiments. We have now added further discussion about these previous findings in our revised manuscript.

There are some issues with the Müller-Lyer experiment. The stimulus is in dot format but there is no image of the test stimulus that can allow readers to judge how much illusion they see in this format.

We apologize unreservedly for this oversight. We uploaded an incorrect version of this figure in our previous submission. Our figure now contains an example of the Mueller-Lyer illusion. We also included a video of the flashing stimuli. We hope the reviewer agrees that this is a convincing version of the Mueller-Lyer illusion.

The methods describe a figure that is only fins with no elements along the space between the two vertices. Eight contextual dots are mentioned, I assume that is 4 on each fin, making 5 dots in total for the fins and vertex on one end and another 5 at the other end, for 10 dots total The authors report a 10% shift in length between the fins out and fins in versions in their behavioral test. This is a small illusion for the Müller-Lyer. In 1970, Coren reported that the illusion effect was about 20% for a line version and 15% for a dot version (with 2 dots to make each fin, so 6 dots in all). Both are larger effects than those reported by the authors here.

We apologize that the stimulus illustration was missing from the previous submission. This has now been included so the reviewer will find it easier to appreciate the illusion in our stimuli.

The perceptual effect was expressed in binary log units, but the reviewer's understanding of the illusion strength is accurate. Converting the mean positions estimated, we obtain an illusion effect of just under 11%. While this is subtly lower than Coren's estimate of 15% for the dot variant, it is not far off. On this note, it is indeed likely that the dot versions of the illusion are weaker than classical versions using solid lines, especially in the absence of an arrow shaft. Please also keep in mind that our stimuli were designed specifically for the fMRI experiment to maximize the cortical territory activated. This means they were larger and positioned at greater eccentricities than stimuli in any psychophysical studies we are aware of. Finally, many psychophysical studies on the Mueller-Lyer illusion used a line-bisection task whereby both the inward and outward fin configuration are part of

the same stimulus, and the observer must choose the “perceived midpoint” of the line. The way the illusion is measured can potentially alter the effect size as well.

Part of that may be due to the measurement of the endpoints rather than some size matching. Gillam and Chambers (1985) report very little Müller-Lyer effect when they had subjects adjusted dots to line up with the endpoint locations. The effect was six times larger when the subjects matched perceived length with an adjustable comparison line. Reduced illusion for the line endings is consistent with the result that reaching to the vertices is quite accurate (Cook, 1979; Thompson & Westwood, 2007).

As already mentioned, the way the illusion is measured may influence the illusion strength. But for the purpose of our study, it is of course crucial to use the same kind of adjustment stimuli as the targets presented in the fMRI experiment. Had we used an adjustable comparison line this would introduce further differences that are impossible to disentangle. The fact that such parameters influence the illusion hint at further factors at play here. Gillam and Chambers’ results are interesting in this context. They showed that adjusting the length of a (spatially displaced) line reference produces a stronger illusion than adjusting the position of dots aligned with the end points of the illusion. However, this finding comes with several caveats: First, the cortical activation evoked by a line inevitably differs from that evoked by dot stimuli. It is possible that the length of the reference line is *also* misestimated due to the blurring of the cortical activity, essentially causing a perceived contraction of the line. In turn, this would result in an exaggerated illusion magnitude. Second, by spatially displacing the reference line observers must compare the shaft lengths across visual space, adding uncertainty compared to judging the spatial alignment of end points. This latter point may be further exacerbated by cognitive factors influencing perceptual decision-making. Such questions should be addressed with more extensive psychophysical experiments.

Importantly, we do not think these points are particularly relevant for the aims of our study. We found a robust illusory effect, even though it may have been weaker than with other stimuli. This effect is consistent with the fMRI effects. It is possible that our experiment taps mostly into the low-level factors (please also see our response to the following comment). Any behavioral effect must necessarily result from the whole brain working in concert, and so may combine several factors. Perhaps a stronger illusion (using solid arrow lines or a different comparison task) incorporates additional factors that do not contribute to our present results. Future fMRI experiments may be able to test if these factors (insofar that they exist) are also represented in V1. We now briefly discuss this broader issue in our revised manuscript.

Finally, the recovered location of the vertex might show an influence of the responses to the contextual dots biasing them inward for the fins in condition and vice versa. This is just the low-pass filter model of the effect but now applied to the cortical patterns of activity. I don't see how the authors have controlled for this. I may have missed something.

No, the reviewer did not miss anything – quite to the contrary, they in fact made our point! We already explicitly argued in our previous manuscript that the shifted activity peak may reflect the low-level processing in V1. We called this the spatial pooling account. As the reviewer described it, it is the low-pass filtering of the cortical activity pattern by receptive fields in V1, and we have now added this phrase for additional clarification in the revised manuscript.

We designed our experiment to single out the response to the flickering target dots by contrasting them against the static contextual dots; however, this does presumably not remove the local interactions between the target and context dots entirely. In the previous manuscript, we also presented several control analyses to account for the potential influence of voxels responding to these contextual dots. This showed that even a strict data selection showed the shifted fMRI responses, but the effect was much reduced. We therefore suggested that low-pass filtering may in fact be how the Muller-Lyer illusion arises. We note that this interpretation is consistent with psychophysical experiments showing that adaptation to low spatial frequencies reduces the strength of the Mueller-Lyer illusion¹⁴. Our results now provide neurophysiological evidence in support of this hypothesis.

However, we now also include data from V2 and V3, extrastriate regions with larger receptive field sizes than V1. A simply low-pass filter of the visual input would predict even greater shifts in activity peaks. Moreover, since V2 and V3 are quadrant-field maps, they have only sparse sampling of the horizontal meridian, where the target dots were located. This means the reconstructions of responses in visual space would be dominated by pRFs away from the horizontal meridian. This predicts reconstructions dominated by the contextual dots rather than the target dots, which also predicts greater activity shifts (because the contextual dots are physically different). Despite these points, if anything we observed a reduce activity shift in V2 and V3 than V1. The data from V1 show the greatest similarity with the observed perceptual effect. Our interpretation is that it is the low-pass filtering in V1 that gives rise to the Muller-Lyer illusion.

As we already alluded to above, any illusory percept must be the combination of multiple processes across the whole brain; it is therefore entirely possible that this low-level process only partially explains the Mueller-Lyer illusion and that different stimulus designs may be able to tap into the

relative contributions of these factors. However, it seems likely that this low-level process plays a large role in most versions of this illusion.

Crucially, our experiments directly compare the Mueller-Lyer to the curveball illusion. We found that the latter is not represented in V1. This suggests that position shifts are not necessarily represented in V1. Taken together with the potential low-level explanation of our Mueller-Lyer results, we surmise that many of the previously reported effects for size illusions in V1 (e.g. ¹⁻⁶) may also have low-level explanation. For some of these effects involvement of feedback from higher regions modulating V1 responses seems likely^{1,6,15} – but either way we can conclude that V1 does not generally encode perceived stimulus size/position.

For the curveball illusion, the absence of an effect in V1 has already been reported (Liu et al, 2019).

We agree, and we already discussed his study in our previous manuscript. We regard Liu et al to be an important study; however, decoding analysis is agnostic to the nature of the underlying representation and the absence of evidence for V1 representation in their study could be due to confounds with that kind of analysis (e.g., noise factors like eye or head movement, experimental choices like the number of voxels to include in the analysis). Our reconstruction approach allows us to explicitly test the hypothesis based on a visual encoding model. We provide convergent evidence that the curveball illusion is not represented in early visual cortex. This is interesting especially in light of previous reports that V1 signals correlated with similar motion-induced position shifts¹¹⁻¹³ (although the inverted sign in some of these findings suggests that the situation is more complex).

References

1. Murray, S. O., Boyaci, H. & Kersten, D. The representation of perceived angular size in human primary visual cortex. *Nat. Neurosci* **9**, 429–434 (2006).
2. Fang, F., Boyaci, H., Kersten, D. & Murray, S. O. Attention-dependent representation of a size illusion in human V1. *Curr. Biol* **18**, 1707–1712 (2008).
3. Ni, A. M., Murray, S. O. & Horwitz, G. D. Object-Centered Shifts of Receptive Field Positions in Monkey Primary Visual Cortex. *Curr. Biol.* (2014)
4. He, D., Mo, C., Wang, Y. & Fang, F. Position shifts of fMRI-based population receptive fields in human visual cortex induced by Ponzo illusion. *Exp Brain Res* (2015)
5. Pooresmaeili, A., Arrighi, R., Biagi, L. & Morrone, M. C. Blood Oxygen Level-Dependent Activation of the Primary Visual Cortex Predicts Size Adaptation Illusion. *J. Neurosci.* **33**, 15999–16008 (2013).
6. Sperandio, I., Chouinard, P. A. & Goodale, M. A. Retinotopic activity in V1 reflects the perceived and not the retinal size of an afterimage. *Nat. Neurosci.* **15**, 540–542 (2012).
7. Schwarzkopf, D. S., Song, C. & Rees, G. The surface area of human V1 predicts the subjective experience of object size. *Nat. Neurosci* **14**, 28–30 (2011).
8. Schwarzkopf, D. S. & Rees, G. Subjective size perception depends on central visual cortical magnification in human v1. *PLoS ONE* **8**, e60550 (2013).
9. Moutsiana, C. *et al.* Cortical idiosyncrasies predict the perception of object size. *Nat Commun* **7**, 12110 (2016).
10. Liu, S., Yu, Q., Tse, P. U. & Cavanagh, P. Neural Correlates of the Conscious Perception of Visual Location Lie Outside Visual Cortex. *Curr Biol* **29**, 4036-4044.e4 (2019).
11. Whitney, D., Westwood, D. A. & Goodale, M. A. The influence of visual motion on fast reaching movements to a stationary object. *Nature* **423**, 869–873 (2003).
12. Whitney, D. & Bressler, D. W. Spatially asymmetric response to moving patterns in the visual cortex: re-examining the local sign hypothesis. *Vision Res.* **47**, 50–59 (2007).
13. Fischer, J., Spotswood, N. & Whitney, D. The emergence of perceived position in the visual system. *J Cogn Neurosci* **23**, 119–136 (2011).
14. Carrasco, M., Figueroa, J. G. & Willen, J. D. A test of the spatial-frequency explanation of the Müller-Lyer illusion. *Perception* **15**, 553–562 (1986).
15. Song, C., Schwarzkopf, D. S. & Rees, G. Interocular induction of illusory size perception. *BMC Neurosci* **12**, 27 (2011).

REVIEWERS' COMMENTS:

Reviewer #1 (Remarks to the Author):

The authors have responded adequately to most of my requests and the manuscript is, in my opinion, significantly improved. Newly added animations of the stimuli greatly aid the understanding of the experiment. I appreciate the inclusion of the results from V2 and V3 in the supplementary material and I think they help to make a stronger paper overall. The authors did not include the analysis of V3A that I suggested and seemingly for good reason. It was certainly not crucial to the point they made.

I have one clarifying question:

The peaks of the pRF maps correspond to a physical location, and the perceived endpoints of the illusion also correspond to a physical location. Why does the correlation between fMRI and perceptual effect use a difference between peaks for the fMRI data, but a log-transformed ratio of differences between illusion endpoints and some other location for the perceptual data? Wouldn't it be more straightforward to use the difference between locations for both?

Reviewer #2 (Remarks to the Author):

This is fine. No further suggestions.

Reviewer #1

“I have one clarifying question: The peaks of the pRF maps correspond to a physical location, and the perceived endpoints of the illusion also correspond to a physical location. Why does the correlation between fMRI and perceptual effect use a difference between peaks for the fMRI data, but a log-transformed ratio of differences between illusion endpoints and some other location for the perceptual data? Wouldn't it be more straightforward to use the difference between locations for both?”

Response: We thank the reviewer for the question and apologise for the confusion. For the Muller-Lyer experiment, we have opted to use log-transformed ratio of differences to be consistent with our previous studies on size perception and to account for Weber's law. The reviewer is correct in that we could also use the difference between locations to calculate the perceptual effect, here we show that this produces almost identical statistical result ($r = 0.52$, compared to $r = 0.51$ when we calculated the perceptual effect using log-transformed ratio of differences).